# An Efficient Computation Offloading Strategy with Mobile Edge Computing for IoT

**DOI:** 10.3390/mi12020204

**Published:** 2021-02-17

**Authors:** Juan Fang, Jiamei Shi, Shuaibing Lu, Mengyuan Zhang, Zhiyuan Ye

**Affiliations:** Faculty of Information Technology, Beijing University of Technology, Beijing 100124, China; shijiamei@emails.bjut.edu.cn (J.S.); mengyuanzhang@emails.bjut.edu.cn (M.Z.); yezy@emails.bjut.edu.cn (Z.Y.)

**Keywords:** mobile edge computing, computation offloading, offloading strategy, genetic algorithm

## Abstract

With the rapidly development of mobile cloud computing (MCC), the Internet of Things (IoT), and artificial intelligence (AI), user equipment (UEs) are facing explosive growth. In order to effectively solve the problem that UEs may face with insufficient capacity when dealing with computationally intensive and delay sensitive applications, we take Mobile Edge Computing (MEC) of the IoT as the starting point and study the computation offloading strategy of UEs. First, we model the application generated by UEs as a directed acyclic graph (DAG) to achieve fine-grained task offloading scheduling, which makes the parallel processing of tasks possible and speeds up the execution efficiency. Then, we propose a multi-population cooperative elite algorithm (MCE-GA) based on the standard genetic algorithm, which can solve the offloading problem for tasks with dependency in MEC to minimize the execution delay and energy consumption of applications. Experimental results show that MCE-GA has better performance compared to the baseline algorithms. To be specific, the overhead reduction by MCE-GA can be up to 72.4%, 38.6%, and 19.3%, respectively, which proves the effectiveness and reliability of MCE-GA.

## 1. Introduction

Mobile cloud computing (MCC), the Internet of Things (IoT), and artificial intelligence (AI) are evolving rapidly, so new applications such as video analysis, virtual reality, and intelligent vehicles are constantly emerging. Moreover, applications may require real time processing such as drone flight control applications, Augmented Reality (AR)/Virtual Reality (VR), and online gaming, with requirements of latencies below a few tens of milliseconds [1]. Cisco notes that by 2021, IoT devices will dominate connectivity. Of the 27.1 billion connected devices in the world, the Internet of Things will reach 13.7 billion [2]. Although MCC can provide centralized computing resources, due to a large amount of data to be transmitted, the transfer of all IoT sensed mobile terminal data to the cloud data center will bring huge crowding pressure and high delay to the network, thus seriously affecting the quality of user experience.

To solve the above problems, mobile edge computing (MEC) emerged. MEC refers to the formation of the edge cloud on the edge servers close to mobile devices in the IoT. It provides users with computing and storage resources which reduces the share of network resources, the energy consumption of UEs, and the network delay [3,4,5]. With the help of MEC, the intelligence of IoT can be improved and the IoT can be rooted in every vertical industry. At present, MEC has attracted extensive attention in the industry. MEC has many application scenarios, among which the typical application scenarios are as follows. (1) There are some local edge application services related to network information function opening and network access, such as positioning and navigation system based on the wireless network [6]. (2) There are some high-definition video acceleration servers. The powerful computing capacity of MEC can greatly reduce the response delay of users while watching videos to ensure the smoothness [7,8]. (3) There are some network edge application services for industrial IoT, Internet of vehicles, and other application scenarios, which have ultrahigh demand for delay, reliability, and computing performance [9,10]. (4) There are many computationally intensive tasks in the blockchain that need to be offloaded to the edges [11,12]. By combining the features of decentralization, consensus mechanism and peer-to-peer interconnection of the block chain technology, the edge nodes can be trusted and authenticated to improve the security of the service and system [13].

MEC is a promising enabler of future Internet such as the fifth generation (5G) [14], enabling multiple future 5G applications and network services such as IoT applications, haptic Internet, AR/VR use cases, or remote driving [15]. Computation-intensive applications, such as AR and VR, require a lot of computation in a very short period of time, which need very high computing power. For example, AR is a technology that uses additional information generated by computer to enhance or extend the user’s view of the real world. Additionally, VR is a computer simulation technology that uses a computer to fuse multiple information and entity behavior to simulate three-dimensional dynamic views. Both of them need to collect real-time information about the status of users including locations and orientations, and then perform calculations. The MEC server can provide rich computing resources and storage resources, buffer audio and video content according to the location information, determine the push content, send it to the user or quickly simulate the 3D dynamic view, and interact with the user, thus greatly improving the user experience.

In MEC, the improvement of mobile application performance is largely dependent on efficient task offloading decisions. Therefore, offloading decision-making has been widely concerned by scholars in recent years. Yang et al. [16] considered the capacity constraints of lead-time and backhaul links and the maximum delay constraints of users and proposed an effective unloading scheme to minimize the total network energy consumption. Zhang et al. [17] proposed a computation offloading scheme for energy perception by weighing energy consumption and time delay, and introduced the residual energy of the smart device battery into the definition of the weighted factor of energy consumption and delay, which effectively reduces the total system consumption. However, the above literature failed to allocate the limited wireless and computing resources reasonably. Tong et al. [18] proposed an adaptive task offloading and resource allocation algorithm in MEC environment. The algorithm used the Deep Reinforcement Learning (DRL) method to determine whether the task needs to be offloaded and allocate computing resources for the task. However, the deep reinforcement learning method also has some defects. It is difficult to adjust parameters, and the training time is long. There are also many solutions for offloading problems under different environmental scenarios from the perspective of optimizing energy consumption. Zhang et al. [19] adopted artificial fish swarm algorithm to design the offloading strategy of energy consumption optimization under the constraint of time delay. This strategy considered the link condition in the task data transmission network and effectively reduced the equipment energy consumption, but it had the defect of high algorithm complexity. In the scenario of multiple resources, Xu et al. [20] designed a task scheduling algorithm of energy consumption minimization particle swarm optimization for multiple resources matching to reduce energy consumption of edge terminal equipment. Zhao et al. [21] proposed a privacy perception computing offloading algorithm based on Lyapunov optimization theory. Liu et al. [22] studied deep learning task offloading. In order to better deploy deep learning applications and optimize network power consumption, a set of sparse beamforming framework based on mixed L1/L2 norms was proposed. However, the above literature does not take into account the delay. At present, some researches regard the computing task of mobile terminal offloading as composed of several subtasks with dependencies among each other and consider the fine-grained task offloading. Ding et al. [23] studied the code-oriented partition computing offload strategy and proposed an offloading strategy to determine the user’s execution location and minimize the system overhead. However, the parallelism of the tasks is not considered.

Different from the above methods, our paper designs an efficient computation offloading strategy in the auxiliary network with multiple MEC servers, which considers the partition of users’ applications. It realizes fine-grained task offloading scheduling. Our study aims at minimizing the overhead of generated applications by improving the parallel computing capacity of MEC servers. The main contributions of this paper are shown as follows.

We consider the heterogenous properties of MEC and the resource limitation of UEs and MEC servers. We jointly optimize the execution delay and energy consumption of applications generated by UEs.We consider fine-grained task computation offloading of fine-grained tasks, which have dependencies, model the user-generated mobile application as a directed acyclic graph, and make the parallel processing of tasks possible.In order to reduce the overhead of applications generated by UEs and improve the utilization rate of system resources, we proposed a multi-population coevolutionary elite genetic algorithm (MCE-GA) to solve resource allocation and task scheduling problem. By simulation experiments, we verify the effectiveness of the MCE-GA algorithm.

The rest of this paper is organized as follows. Section 2 describes the system model and formulates the problem of computing offloading. Section 3 describes the proposed computation offloading algorithm. Section 4 describes the proposed simulation analysis method and compares it with other algorithms. Section 5 summarizes our study.

## 2. System and Computation Model

In this section, we describe the system and computation model of MEC in detail.

### 2.1. System Model

We consider the scenario of multiple users and multiple MEC servers. In the whole network, UEs are connected through base stations and wireless channels. Multiple MEC servers are deployed beside the base stations to provide computing services for UEs. Cloud servers are located on top of the core network far from the UEs. Compared with the unpredictable delay and long transmission distance caused by mobile cloud computing technology used by UEs to offload computing to cloud servers, MEC can provide computing services for UEs more quickly and efficiently and relieve the pressure on core networks [24]. In our work, we only consider the computation offloading problem between the user layer and the edge layer, where the user chooses to offload the task to the local or MEC server.

The overall network of system is shown in Figure 1. In this system, the top end is the cloud server communicating with the base station through a switch. There are multiple base stations in the whole network. A number of small MEC servers are deployed near each base station, and the communication can be conducted between MEC servers. On the terminal smart device side, the tasks can be executed directly on the local server or sent via the data transfer unit to a MEC server in its area for remote computation.

The whole network of system is composed of several small areas, which are independent of each other. We consider the computation offloading situation in a small area and have the following assumptions. There are U UEs and M MEC servers in an area. $UE_{i}$ (i∈{1,2,…,U}) offloads computing tasks to $MEC_{j}$ (j∈{1,2,…j,…M}) through wireless links communication. The wireless links are orthogonal, so links do not interfere with each other. What’s more, $UE_{i}$ communicates with all MEC servers that can be connected and perceives the computing resources of these MEC servers. $UE_{i}$ can choose to compute locally or offload MEC servers in this area. Let $f_{i}$ and $F_{j}$ denote the computing capacity of $UE_{i}$ and $MEC_{j}$, respectively. The main notations involved in this paper are given in Table 1.

### 2.2. Application Model

We assume that the application generated by UE is composed of several tasks with dependency. We first model it using a directed acyclic graph (DAG) to achieve fine-grained task offloading scheduling. Based on that, we analyze the probability of the parallel processing for tasks, which can speed up the execution efficiency and corresponds to a more realistic scenario. Figure 2 shows an example of the application model produced by UE.

We model the application as a DAG graph, where $UE_{i}$ generates application $A_{i}$. So $UE_{i}$ can also be expressed as $A_{i}=\left\{V_{i},E_{i}\right\}$, where $V_{i}$ is the set of generated tasks, and $E_{i}$ is the dependency relationship between tasks. We define a task generated by UE as $v_{k}$ where $v_{k}\in V_{i}$. For example, in Figure 2, there are $v_{0}$,$\;v_{1}\in E_{i}$, and there is a directed edge from $v_{0}$ to $v_{1}$ in $A_{i}$, thus $v_{1}$ must be executed after $v_{0}$. Moreover, we define $v_{k}=\left\{w_{k},d_{k},\sigma_{k},\tau_{k}\right\}$.$\;w_{k}$ refers to the total workload of $v_{k}$, which indicates the number of CPU clocks cycles required to execute $v_{k}$.$\;d_{k}$ indicates data size of the task and $\sigma_{k}$ refers to the ratio of the output/input data sizes of tasks generated by $UE_{i}$,$\;\tau_{k}$ presents the maximum delay a task can tolerate. We use a binary variable $I_{k,i,j}\in \left\{0,1\right\}$ to indicate whether the task was offloaded to the $MEC_{j}$. We use $I_{k,i,0}=1$ to indicate that the task was executed locally. Since we assume one task can only be executed on one position, we set $\sum_{j=0}^{M}I_{k,i,j}=1$.

### 2.3. Communication Model

In the whole system, there are mainly three kinds of communication depending on user’s offloading location. (1) $UE_{i}$ offloads $v_{k}$ to $MEC_{j}$. (2) $MEC_{j}$ receives the computation result of $v_{k}$ from $MEC_{j^{\prime }}$. (3) $UE_{i}$ receives the result of $v_{k}$ from $MEC_{j}$.

According to Rayleigh fading channel model [25], the rate of $UE_{i}$ to transmit $v_{k}$ to $MEC_{j}$ can be defined
(1)$$r_{i,j,v}=Blog\left(1+\frac{p_{i,j}h_{i,j}}{d_{i,j}^{\theta }N}\right),$$
where B represents the transmission bandwidth, $p_{i,j}$ represents the transmission power from $UE_{i}$ to $MEC_{j}$, $h_{i,j}$ represents the channel gain between $UE_{i}$ and $MEC_{j}$, $d_{i,j}$ represents the distance between $UE_{i}$ and $MEC_{j}$, θ represents the path loss exponent, and N represents Gaussian noise.

The rate of $UE_{i}$ to receive the result of $v_{k}$ from $MEC_{j}$ can be defined as
(2)$$r_{j,i,res}=Blog\left(1+\frac{p_{j,i}h_{j,i}}{d_{j,i}^{\theta }N}\right).$$

Similarly, $p_{j,i}$ represents the receiving power of $UE_{i}$ from $MEC_{j}$, $h_{j,i}$ represents channel gain, and $d_{j,i}$ represents the distance between $MEC_{j}$ and $UE_{i}$.

Moreover, when $MEC_{j}$ executes task $v_{k}$, if $v_{k-1}$ is executed in $MEC_{j^{\prime }}$, then the transmission rate of receiving the result of the precursor task from $MEC_{j^{\prime }}$ is defined as
(3)$$r_{j^{\prime },j,pre}=Blog\left(1+\frac{p_{j^{\prime },j}h_{j^{\prime },j}}{d_{j^{\prime },j}^{\theta }N}\right),\;$$
where $p_{j^{\prime },j}$ represents transmission power of $MEC_{j^{\prime }}$, $h_{j^{\prime },j}$ represents channel gain, and $d_{j^{\prime },j}$ represents distance between $MEC_{j^{\prime }}$ and $MEC_{j}$.

### 2.4. Computation Model

In this subsection, we describe the computation model in detail. Computing offloading is performed either locally or by MEC servers. We consider fine-grained task offloading, so the dependencies between tasks are also needed to be considered. The dependencies between tasks mean that all precursors of task must have finished executing before it can begin executing. Obviously, more complex dependencies between tasks will increase the complexity of task computation offloading decision. Next, we give the definition of ready time as follows.

Definition 1 (Ready Time). Ready Time denotes the total wait time required for task $v_{k}$ to start execution.

(1)Local Computation Model

According to the task model defined above, $w_{k}$ is the workload of the task $v_{k}$, which is the total CPU cycles needed to execute the task. $f_{i}$ is the computing ability of $UE_{i}$. The local execution delay of task $v_{k}$ for $UE_{i}$ is defined as
(4)$$T_{k,i,exe}^{l}=\frac{w_{k}}{f_{i}}.$$

In addition, we assume that $UE_{i}$ have limited resources and can only execute one task at a time, so $UE_{i}$ have a waiting queue to store tasks that need to be computed locally. A wait delay occurs when multiple tasks of $UE_{i}$ are offload locally. The waiting delay of task $v_{k}$ is the total delay of the completion delay of the tasks in the waiting queue that arrived earlier than task $v_{k}$. 

We use $FT_{pre,i}$ to represent the completion delay of the precursor of task $v_{k}$, $FT_{pre,i}^{res}$ to represent the return delay after the precursor for task $v_{k}$ was completed, and $T_{wait}^{l}$ to represent the local execution delay of task $v_{k}$. $FT_{pre,i}^{res}$ can be defined as
(5)$$FT_{pre,i}^{res}=\frac{d_{pre}*\sigma_{pre}}{r_{j,i,pre}},$$
where $d_{pre}$ represents data size of the precursor of task $v_{k}$, $\sigma_{pre}$ represents ratio of the output data size to the input data size of task $v_{pre}$, and $r_{j,i,pre}$ represents the rate from $MEC_{j}$ to $UE_{i}$.

Thus, the ready time of $v_{k}$ can be defined as
(6)$$RT_{k,i}^{l}=\underset{pre\in pred\left(v_{k}\right)}{\max}(FT_{pre,i}+FT_{pre,i}^{res},\;T_{wait}^{l}),$$
where $T_{wait}^{l}$ indicates the total waiting time required for task $v_{k}$ to start execution.

Therefore, the total delay for task $v_{k}$ of $UE_{i}$ to execute locally can be defined as
(7)$$FT_{k,i}^{l}=T_{k,i,exe}^{l}+RT_{k,i}^{l}.$$

According to [26], the total energy consumption for task $v_{k}$ of $UE_{i}$ to execute locally can be defined as
(8)$$E_{k,i}^{l}=k_{i}w_{i}f_{i}^{2},$$
where $k_{i}$ is the coefficient factor of $UE_{i}$’s chip architecture.


(2)MEC Server Computation Model


When $UE_{i}$ chooses to offload to MEC server for computation, the transmitting delay of task $v_{k}$ from $UE_{i}$ to $MEC_{j}$ is defined as
(9)$$T_{k,i,trs}^{j}=\frac{d_{k}}{r_{i,j,k}}.$$

The computational time of task $v_{k}$ executed at $MEC_{j}$ is defined as
(10)$$T_{k,i,exe}^{j}=\frac{w_{k}}{F_{j}},$$
where $d_{k}$ is the data size of $v_{k}$,$\;r_{i,j,k}$ is the transmission rate from $UE_{i}$ to $MEC_{j}$, and $F_{j}$ is the computing ability of $MEC_{j}$.

Similarly, the ready time for the task $v_{k}$ needs to be considered. When the task $v_{k}$ chooses to offload on $MEC_{j}$, the ready time involves waiting for the precursor tasks of task $v_{k}$ to finish executing and passing the result back to $MEC_{j}$, waiting for the transmitting delay of task $v_{k}$ from $UE_{i}$ to $MEC_{j}$. In addition, we suppose that the MECs also have limited computing resources and execute one task at a time, so the ready time also includes the sum of the completion delays of tasks in the $MEC_{j}$ waiting queue that arrive before the current task. Figure 3 shows an example of the task queue model.

Thus, the ready time for task $v_{k}$ executed on $MEC_{j}$ is expressed as $RT_{k,i}^{j}$ and can be defined
(11)$$RT_{k,i}^{j}=\underset{pre\in pred\left(v_{k}\right)}{\max}(FT_{pre,i}+FT_{pre,i}^{res},\;T_{wait}^{j},\;T_{k,i,trs}^{j}),$$
where $FT_{pre,i}$ is the delay of the precursor of $v_{k}$, $FT_{pre,i}^{res}$ is the result return time of the precursor task of $v_{k}$, and $T_{wait}^{j}$ is the queue wait delay in $MEC_{j}$.

Thus, the total delay of task $v_{k}$ executed at $MEC_{j}$ can be defined as
(12)$$FT_{k,i}^{j}=T_{k,i,exe}^{j}+RT_{k,i}^{j}.$$

The total energy consumption of task $v_{k}$ executed at $MEC_{j}$ can be defined as
(13)$$E_{k,i}^{j}=p_{i,j}*T_{k,i,trs}^{j}+p_{i,j}*T_{k,i,res}^{j}.$$

### 2.5. Problem Formulation

In this subsection, we formulate the computation offloading problem to optimize the overhead of applications generated by $UE_{i}$.

The total delay of $UE_{i}$ can be formulated as
(14)$$T_{i,total}=\overset{\left|v_{i}\right|}{\underset{k=1}{\sum }}(\left(1-\overset{M}{\underset{j=1}{\sum }}I_{k,i,j}\right)FT_{k,i}^{l}+\overset{M}{\underset{j=1}{\sum }}I_{k,i,j}FT_{k,i}^{j}).$$

The total energy of $UE_{i}$ can be formulated as
(15)$$E_{i,total}=\overset{\left|v_{i}\right|}{\underset{k=1}{\sum }}(\left(1-\overset{M}{\underset{j=1}{\sum }}I_{k,i,j}\right)E_{k,i}^{l}+\overset{M}{\underset{j=1}{\sum }}I_{k,i,j}E_{k,i}^{j}).$$

Some tasks require low latency and some require low energy consumption. In order to consider these two consumptions, we draw on the work of [27] and define the total overhead of $UE_{i}$ as
(16)$$C_{i}=\beta *E_{i,total}+\left(1-\beta \right)*T_{i,total},$$
where β∈[0,1] represents the tradeoff parameter of delay and energy. 

In order to minimize the execution overhead of the application generated by $UE_{i}$, for $UE_{i}$, we can formulate the problem as
(17)$$\begin{array}{cc} P1: & \underset{I_{i}}{\min}C_{i}\\ s.t. & C1:T_{i,total}\leq \overset{\left|v_{i}\right|}{\underset{k=1}{\sum }}\tau_{k}\\ C2: & \left(v_{k},v_{k+1}\right)\in E_{i},\;v_{k},v_{k+1}\in V_{i}\\ C3: & \overset{M}{\underset{j=0}{\sum }}I_{k,i,j}=1,I_{k,i,j}\in \left\{0,1\right\} \end{array}.$$
where $I_{i}$ is the offloading decision set of $UE_{i}$. In P1, C1 is the execution delay constraint of the application generated by $UE_{i}$. C2 indicates the sequence of task execution the dependency between the tasks. C3 represents the decision of task offloading. We limit that a task can only be executed at one location at a time. 

## 3. Proposed Algorithm

In this paper, we propose a Multi-population Coevolutionary Elite Genetic algorithm (MCE-GA) to solve this problem and make a reasonable offloading decision. We introduce multiple populations for parallel optimization. We endow different populations with different controls to achieve different search objectives. Each population evolves in parallel according to its own different evolutionary strategies and genetic manipulation. By co-evolution of migration operator, the optimal solution is obtained. Artificial selection operator is used to save the optimal individual and the convergence of the algorithm is judged in each evolution.

### 3.1. The Flow of MCE-GA

In this subsection, we introduce the flow of MCE-GA.


(1)Chromosome and Fitness Function


For simplicity, real number coding is used. In our algorithm, individuals are defined as chromosomes, indicating that each chromosome individual is a computation offloading decision to problem P1. We assume that $UE_{i}$ has k tasks. The individual structure of chromosome is shown in Figure 4. Each chromosome individual contains the user’s tasks scheduling policy. The resulting decision is variable for the task, i.e., choosing where the task offloading.

We take the total overhead incurred by $UE_{i}$ as the fitness value of the individual. Thus, the fitness function is defined as
(18)$$Fitness=\beta *E_{i,total}+\left(1-\beta \right)*T_{i,total},$$
where $E_{i,total}$ is the total energy of $UE_{i}$ and $T_{i,total}$ is the total delay of $UE_{i}$.


(2)Initialization and Selection


We generate the initial populations randomly, but within the restraints in Problem P1. The individuals of the populations are generated and the variable values of genes are guaranteed to be within their range. Thus, we set the genes of the initial populations to $a_{i}\left(0\right)=random\left(M\right)$, where random(value) returns a random number in the interval [0,value]. If $a_{i}==j$, then the task $v_{i}$ is executed on $MEC_{j}$. We use $a_{i}=0$ to represent the task $v_{i}$ to be executed locally.

The selection operation is then performed within each population. For selection operations, two methods are used in a wide range, namely, roulette and tournaments. The roulette selection method calculates the probability of each individual according to the fitness value of the individual and randomly selects individuals to form the offspring population according to the probability. Therefore, the greater the fitness value of the individual, the greater the chance it has to be selected. Multiple rounds of selection are required to select mating individuals. A uniform random number between 0 and 1 is generated for each round, and this random number is used as the selection pointer to determine the selected individual. Due to random operation, the selection error of this selection method is also relatively large, and sometimes, the high fitness of the individual is not selected. In addition, the selection strategy roulette used in roulette is more suitable for the maximization problem. Therefore, a tournament selection strategy is adopted in this paper to select a certain number of individuals from the population at a time, and then select the best one to enter the offspring population. The first reason is that this method has a low computational cost. Another point is that choosing good parents is better than choosing only the best parents [28]. This method is more random and has a greater random error. It maintains diversity and has the chance to produce better individuals. Additionally, this method does not require the individual fitness value to be positive or negative.

Therefore, different populations take the selection operation independently. N individuals were randomly selected from the population for fitness comparison, and the highest fitness individuals were inherited to the next generation.


(3)Crossover and Mutation Operation


In order to preserve the diversity of the population and better solve the problem of offloading decision, MCE-GA performs iteration, crossover operation, and mutation operation on each population. Appropriate individuals will be selected for crossover and mutation operation. Then, offspring will be produced.

For crossover operation, we adopted a two-point crossover. Two-point crossover means that two intersection points are randomly set in the individual and then partial gene exchange is carried out. Figure 5 shows the example of the crossover operation for the computation offloading decision variables. First, two intersection points are randomly set in two individuals, and then, part of the chromosomes of two individuals between the two set intersections is exchanged. 

Moreover, when setting the parameters of cross recombination, we generate the recombination probability list and assign different recombination probabilities to different populations. We can maintain the diversity of the population better and avoid local optimality.

For mutation operation, a list of variation probabilities is generated and different variation probabilities are assigned to different populations. An individual is chosen and the mutation probability of each element in the chromosome is in the range of the variables. 

In the case of mutation operation, when the variable value of the chromosome exceeds the range set by the individual chromosome variable, it is repaired. In this paper, truncation repair is adopted, i.e., the nearest boundary value of the element beyond the boundary range is taken.


(4)Migration


By setting different probability values and performing individual migration among populations according to the probability, the diversity of each population is maintained. All populations can share excellent genes, complete their own evolution, and finally, get the optimal solution. When making individual out, we choose preferred emigrated from individuals, and at the time of the move in individual, population with the method of choosing a bad replacement, so that we can speed up the convergence of the algorithm. In the Internet environment, most of the request to delay requirement is high, so offloading decisions need to be made quickly, so we use an elite reserved strategy.


(5)Elite Population Selection


The optimal individual of each evolutionary generation is preserved and compared with the optimal value of the previous generation, and the difference value is taken as the basis to judge the convergence of MCE-GA. For the elite population, we do not take crossover and mutation operation and only retain it.

### 3.2. Offloading Strategy Based on MCE-GA

In this subsection, we introduce our proposed MCE-GA algorithm in detail. The algorithm aims to solve resource allocation and task scheduling problem, as shown in Algorithm 1.

In Algorithm 1, first, we initialize multiple populations and some important parameters, which including evolutionary stagnation threshold, the maximum number of iterations, interval steps value of population migration, mutation operator probability, and crossover operator probability for each population. Moreover, we need to initialize a list to keep track of the elite population. Then, we calculate the fitness value (lines 3–5). The optimal individual in each population is updated to the elite population. The algorithm ends on the basis that the optimal individual in the elite population remains above the specified algebra. When the iteration does not end, the evolutionary operation continues. In the multi-population evolutionary operation, we assume that populations are independent of each other in selection, crossover, and mutation operation. So, for each population, we evolve individuals according to the selection, crossover, and mutation operations described above, and get a new generation of population (lines 8–14). We control population connections and coevolution through population migration. Therefore, the optimal individuals in a population are transferred to other populations through migration operations (lines 15–16). At each iteration, the elite population is renewed and the elite individuals of each population are retained (line 17). When evolution reaches the convergence standard, the value of the best individual is the user’s offloading strategy. The fitness of the individual is the user’s overhead.
**Algorithm 1:** MCE-GA**Inputs:** Population size list PopNum, evolutionary stagnation threshold ε, the iterations T, interval steps of migration migFr, mutation probability list pm, crossover probability list pc**Outputs:**Optimal offloading policy A, the total overhead C1: Randomly initialize the populations Pop and Elite population Elite_pop
2: Initialize the inputs3: **For**
*i* = 1 to PopNum.size
**do**4: Pop = Pop[i];5: Evaluate the fitness value of each individual in the i-th
Pop;6: Update Elite population Elite_pop;7: **While** stopping criterion is not met **do**8: **For**
*i* = 1 to PopNum.size
**do**9:  offspring = Select (Pop[i]);10:  pop = Cross and Mutate (offspring); 11:  Pop[i] = Pop[i]+ pop;12:  evaluate the fitness value of each individual in Pop[i];13:  Select individuals to get a new generation of population;14: **End For**15: **IF** evolutionary algebra % migFr==0
**do**16:  Carry out population migration;17: Update Elite population Elite_pop;18: Return optimal offloading policy and the total overhead.19: **End**

## 4. Simulation and Result

### 4.1. Simulation Setting

We consider an area of 500×500 m^2^. We set the maximum latency of each UE up to 85% of the local latency. Referring to [24,29], we assume our simulation setting parameters. We list the important simulation parameters in Table 2.

In this paper, we consider the three baseline algorithms to evaluate the performance, and make a comparative analysis of their experimental results.

Random selection algorithm (Random): The tasks of applications randomly select offloading locations.

Greedy algorithm (Greedy): The tasks of applications select the best offloading locations in the current situation. Greedy is to achieve global optimization through local optimization, and to construct the optimal solution step by step. At each stage, a seemingly optimal decision is made. Once the decision is made, it will not be changed. Therefore, with the greedy algorithm, the current task of application always chooses what looks like the best offloading location based on its predecessor task processing and the local user and MEC server situation, but ignores its successor tasks. Standard genetic algorithm (SGA): SGA introduces the biological evolutionary principle of “survival of the fittest, survival of the fittest” into the coded tandem population formed by optimized parameters. SGA starts with an initial set of random offloading strategy and optimizes the offloading strategy through some standard genetic operations such as selection operation, crossover operation, and mutation operation until reaching a better offloading strategy or convergence. Because the optimization does not depend on the gradient, it has strong robustness and global search ability. However, immature convergence is indeed a phenomenon that cannot be ignored in the standard genetic algorithm. It mainly shows that all individuals are in the same state and stop evolving. By comparing the proposed MCE-GA algorithm with these three algorithms, the effectiveness of MCE-GA is verified.

### 4.2. Convergence Analysis

In this subsection, the convergence of SGA and MCE-GA are analyzed. The convergence of genetic algorithms has been demonstrated in literature [30,31].

Figure 6 shows the convergence situation of MCE-GA and SGA. We consider that the MEC servers are lack of computing resource and mobile terminals have low latency requirements, so the iteration is set to 150 times in this work. The black line shows the average fitness value of the population, and red line shows the optimal fitness value of the population. In this paper, the fitness value represents the overhead of the UE when computing tasks, so the lower the fitness value, the lower the overhead, and the better the strategy. We can see that MCE-GA basically reaches convergence and maintains stability quickly, and the average fitness value is basically consistent with the best fitness value. However, the overhead of SGA still fluctuates in a small range which has not reached convergence.

Figure 7 shows the different convergence of SGA and MCE-GA. We can see that SGA is easy to fall into “premature” and converge to the local optimal solution. However, MCE-GA algorithm introduces multiple population optimization search at the same time. To achieve the purpose of different searches, different populations are endowed with different control parameters. The introduction of migration operator is to connect multiple populations and realize the co-evolution, which makes MCE-GA obtain the optimal solution. Thus, the overhead of MCE-GA is better than SGA.

### 4.3. Performance Analysis

In this subsection, we analyze the performance of MCE-GA.

Figure 8 shows the impact of different number settings of MEC server on device overhead. The abscissa is the number of MEC servers, and the ordinate is the device overhead. The number of MEC servers is set at 3–7. As the number of MEC servers increases, the overhead of application generated by UE decreases. This is because, the more the MEC servers there are, the more the rich computing resources there are in the system; so, users can greatly reduce the delay in executing tasks, thus reducing the overall overhead. However, we find that when the number of MECs grows from 5 to 7, the overhead is reduced but not by much. This is because, when the number of MECs increases to a certain extent, MEC server resources is particularly rich, where the offloading location is not significantly affected in this case. Therefore, how to set the number of MEC servers to achieve optimal system performance is particularly important. We set the number of MECs to 7, and by comparing the four algorithms, we can see that MCE-GA algorithm can better reduce the overhead of devices. Compared with random method, greedy method, and standard genetic algorithm, the optimization rate of MCE-GA method is 55.5%, 49%, and 7.1%, respectively. Since the random method is to randomly select the unloading location, the overhead incurred is random and expensive. The greedy strategy is to select the local optimal unloading location for the current subtask, but not to consider the offloading location globally, so the overhead is large. The standard genetic algorithm and MCEGA algorithm take into account the global offloading strategy, so that the overhead is small, but the standard genetic algorithm is easy to fall into premature. The offloading decision made by MCE-GA is better. MCE-GA can better reduce the overhead.

Figure 9 shows the overhead of UE with different offloading strategies at different data sizes. From Figure 8, we find that it is better to set the number of servers to 5, because when the number of servers increases to 7, the reduction in overhead is not obvious, but the cost of adding MEC servers increases. Therefore, number of MEC servers is set to 5. Then, we set the data size generated by each task, at 300–1000 KB. The experimental result figure shows that with the increase in data volume, the total overhead of the system will increase. This is because the increase in data volume will consume more computing resources of UE and MEC servers, resulting in an increase in the overall system overhead. When the amount of data is 300 KB, we find that there is little difference between the greedy strategy, the standard genetic algorithm, and the MCE-GA algorithm. Because the amount of data that needs to be computed is small and the computing resources in the system is rich, the overall cost does not differ much. However, the random strategy is a little big; because the task dependence itself is a very complex problem, the improper selection of the strategy will cause a lot of ready delay, which will lead to extremely high overhead. As the data size increases, the advantages of MCE-GA algorithm are reflected, because when the data volume is larger, the resources of MEC server are limited, the choice of offloading strategy becomes particularly important. A good strategy can make the system produce lower overhead. Compared with the random method, greedy method, and standard genetic algorithm, the optimization rate of MCE-GA method was 77.2%, 41.3%, and 25.2%, respectively, when the data volume was 500 KB. The offloading decision made by MCE-GA is better.

Figure 10 shows the total device overhead when multiple UEs perform task computation. We set the number of devices from 10 to 50 randomly. The number of tasks generated by UEs is random, and the data size of each task is random. We can find that as the number of UEs increases, the overall system overhead is on the rise. Additionally, MCE-GA has more advantages. As the more UEs there are, the more computing resources system need. The computing resources of MEC servers and UEs are not infinite, so how to allocate resources and how to offload intelligent devices are particularly important. In terms of offloading strategy selection, MCE-GA has a greater advantage. When the number of equipment is 20, we analyze the performance of the algorithm. Compared with random method, greedy method, and standard genetic algorithm, the optimization rate is 72.4%, 38.6%, and 19.3%, respectively. The proposed algorithm in this paper can make a better offloading decision and reduce overhead of UEs.

## 5. Conclusions

In this paper, we propose the problem of task partitioning computation offloading strategy under multi-UE and multi-MEC model with limited resources. First, we consider fine-grained task computation offloading of fine-grained tasks, which have dependencies, model the user-generated mobile application as a directed acyclic graph, and make the parallel processing of tasks possible. Second, in order to meet low energy consumption and low latency service requirements, we consider the execution delay and energy consumption of applications generated by UEs, and formulate a problem of minimizing overhead of UEs. Then, we propose MCE-GA algorithm for joint optimization. Finally, we study the convergence and performance of MCE-GA by simulation. Simulation results show that the proposed algorithm can find a reasonable allocation of resources and reduce the overhead of the whole system. It is superior to Random, Greedy, and SGA in reducing overhead of UEs. 

For future work, we will take the cloud into consideration and consider the architecture of edge-cloud collaboration. Moreover, we will continue our research by considering user mobility and dynamic computation offloading. Mobility is an inherent feature of many emerging applications, such as AR to assist museum visits to enhance the visitor experience. Therefore, mobility management is a key problem that needs to be solved urgently in computing offloading, and it is also one of the major challenges to further design efficient computing offloading schemes.

## Figures and Tables

**Figure 1 micromachines-12-00204-f001:**
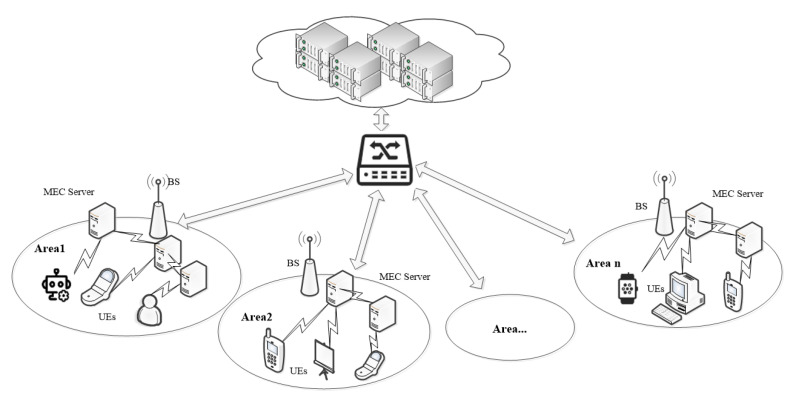
Overall network of system.

**Figure 2 micromachines-12-00204-f002:**
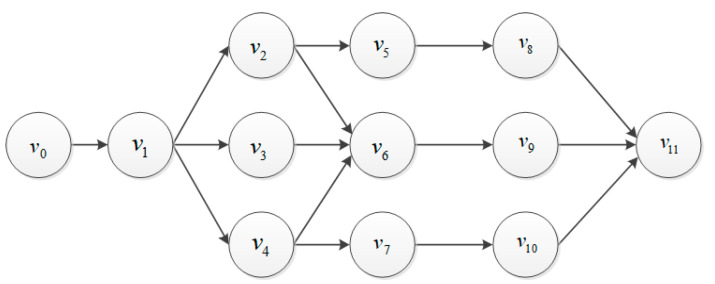
The application model generated by user equipment (UE).

**Figure 3 micromachines-12-00204-f003:**
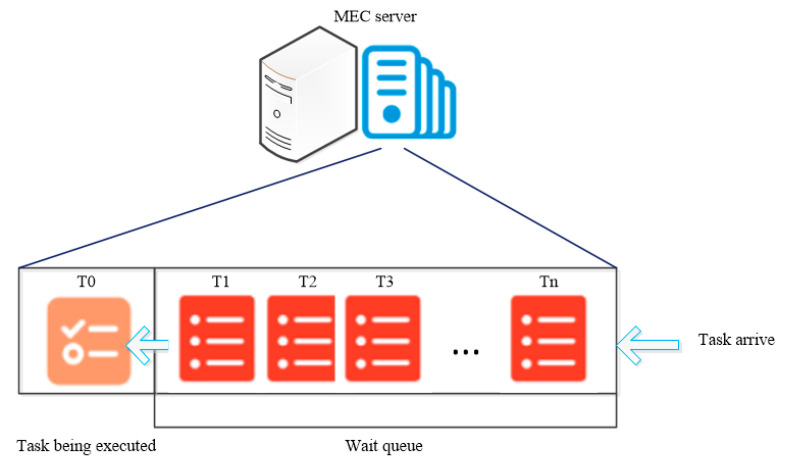
An example of the Task Queue model.

**Figure 4 micromachines-12-00204-f004:**
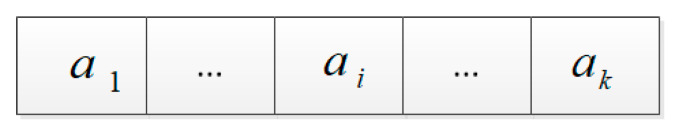
The chromosome structure.

**Figure 5 micromachines-12-00204-f005:**
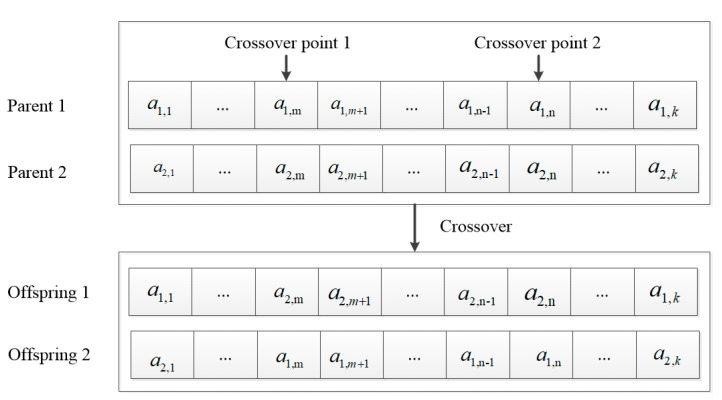
Crossover of the first row of chromosome.

**Figure 6 micromachines-12-00204-f006:**
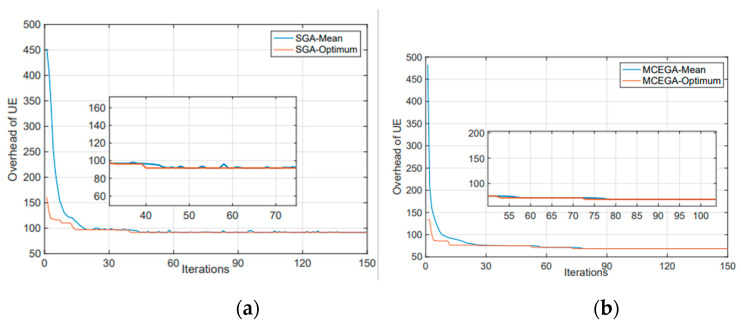
The convergence situation of MCEGA (**b**) and Standard genetic algorithm (SGA) (**a**).

**Figure 7 micromachines-12-00204-f007:**
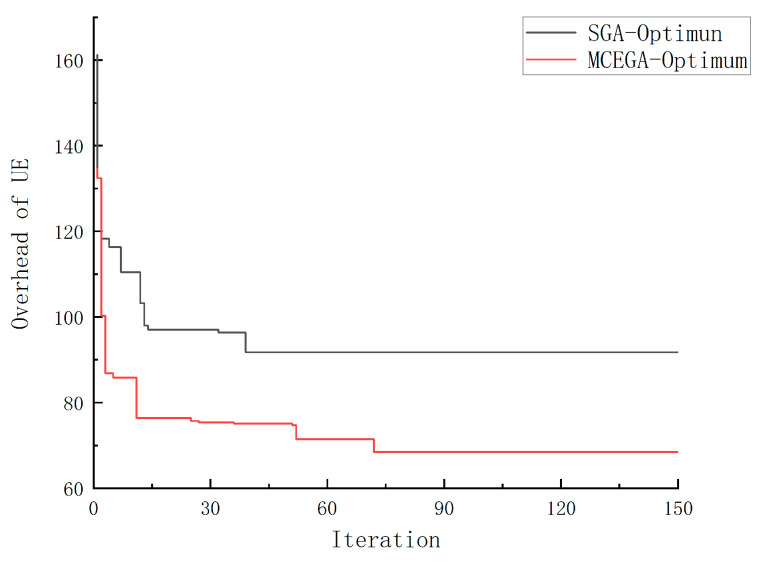
Convergence comparison between MCE-GA and SGA.

**Figure 8 micromachines-12-00204-f008:**
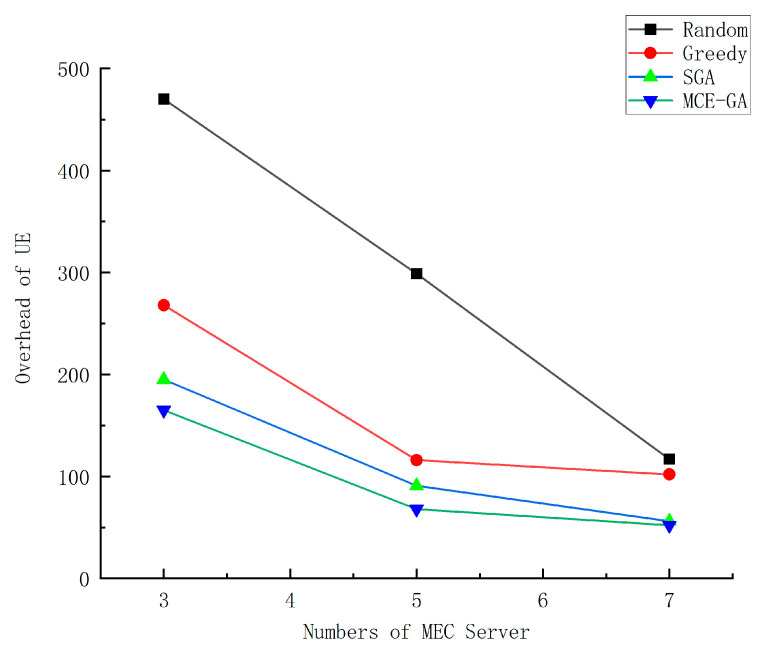
The execution overhead of different number of MEC server.

**Figure 9 micromachines-12-00204-f009:**
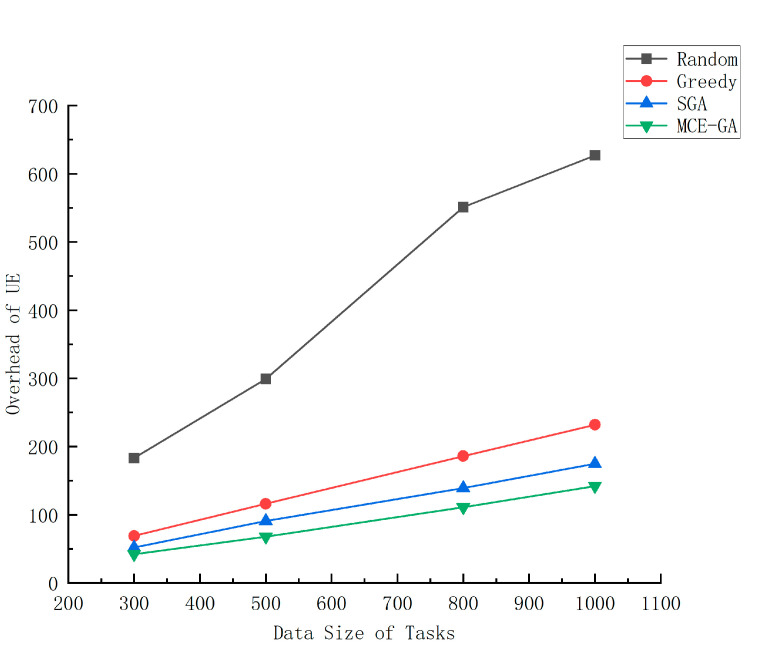
The execution overhead of different data size of tasks.

**Figure 10 micromachines-12-00204-f010:**
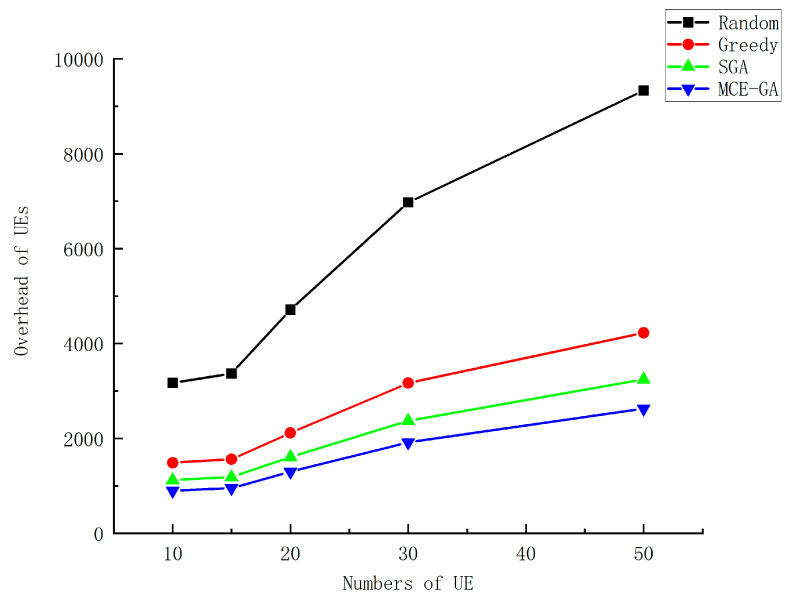
The execution overhead of different numbers of device.

**Table 1 micromachines-12-00204-t001:** Notations.

Symbol	Description
UE	User equipment
MEC	MEC servers
$f_{i}$	Computing capacity of $UE_{i}$
$F_{j}$	Computing capacity of $MEC_{j}$
$A_{i}$	Application generated by $UE_{i}$
$v_{k}$	Task of Application
$w_{k}$	Workload of $v_{k}$
$d_{k}$	Data size of $v_{k}$
$\sigma_{k}$	The ratio of the output/input data size
$\tau_{k}$	Maximum delay a task can tolerate
B	Transmission bandwidth
N	Background noise
p	The transmission power of $UE_{i}$
$d_{i,j}$	The distance between $UE_{i}$ and $MEC_{j}$
$d_{j^{\prime },j}$	The distance of MEC servers
h	The channel gain
$k_{i}$	The coefficient factor of $UE_{i}’s$ chip architecture
θ	The channel fading coefficient

**Table 2 micromachines-12-00204-t002:** The simulation parameters.

Simulation Parameters	Value
Channel bandwidth	180 kHz
Path loss exponent	3
Background noise	$10^{-13}$
Number of tasks	61,223
Data size of tasks	300~1000 kb
Transmission power of UE	3 W
Computation capacity of MEC server	5 GHz
Computation capacity of UE	0.5–1 GHz
Channel fading coefficient	$10^{-3}$
Tradeoff parameter *β*	0.5
Processing density of UE	500~800 cycle/bit
Distance between UE and MEC server	80~100 m
Distance of MEC servers	50~100 m
Coefficient factor of device’s chip architecture	$10^{-20}$
Number of MECs	3–7
Ratio of the output data size to the input data size	0.001~0.005
Interval steps of population migration	5
Size of the populations	10,152,025

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
