# Peer review of "An Efficient Computation Offloading Strategy with Mobile Edge Computing for IoT"

_micromachines, 2021, doi:10.3390/mi12020204_

Round 1
Reviewer 1 Report
Dear Authors,
A reference here need to be added line 114 to 118 sentence: "Compared with
the unpredictable delay and long transmission distance caused by mobile cloud computing technology used by UEs to offload computing to cloud servers, MEC can provide computing services for UEs more quickly and efficiently and relieve the pressure on core networks."
"There are ? UEs and ? MEC servers in a region" --> Region or Area, please be consistent in case that are the same.
There is no definition of the tasks, Authors are assuming the tasks are homogeneus or heterogeneus? I see that task have some dependecies but i didn't see nothing regarding the time per task. Could you explain that with more details?
Sometimes in MEC infrastructures the edge can contain more that one computer, could Authors explain with more details how they do the modeling when tasks are more than the cpus or vcpus in the EDGE?
line 238 replace cinsider by consider.
Figures 6a and 6b will be more interesting for the user if there is a zoom in one corner. https://www.mathworks.com/matlabcentral/fileexchange/59857-zoomplot
Figures 6 to 9 the overhead is %? time?, Put the units in Y axis
Line 383 replace optimalfitness by optimal fitness
Will be amazing if figure 8 have the same colors of figure 9 and 10? Will be more intuitive for the reader to compare and understand what authors are doing.
Could Authors explain with more detail the four strategies? For the reader will be amazing to have a section explaining with detail the four strategies and compare among them in terms of definitions.
Authors are claiming low energy consumtion but there are not results in order to validate this. I understand the overhead of the UE is afecting this values, Could authors provide and explain some "divided" results of this parameters?
Reviewer 2 Report
This paper tries to make a case of task partitioning with computation offloading strategy in multi MEC environment with limited resources. The topic is timely, however, there are some unanswerable questions that needs to be addressed before the final version. I encourage the authors to consider the following major comments.
- First of all, the abstract needs to be revised. The abstract should be highlighting the main findings of the proposed solution, and reporting the numbers promoting proposed approach. Authors need to highlight the performance results; for instance, experimental results show better performance compared to the state-of- the art protocol in terms of xyz etc ...
- The reviewer is afraid that the authors are lacking the updated literature about MEC. The term MEC was coined by ETSI in 2014. While the term specifically referred to mobile networks initially, now ETSI’s scope covers both mobile and fixed networks. The MEC acronym no longer refers to ‘‘Mobile Edge Computing’’ and instead stands for ‘‘Multiple-access Edge Computing’’. The extension includes non-cellular technologies and now Wi-Fi is included within MEC’s scope. Authors are referred to the wealth of material on Edge/Fog computing especially by Mung Chiang from Princeton and OpenFog Consortium. There is a lot of material out there about Edge computing and should be checked.
- In Section 2.1 authors have claimed that “Therefore, we only consider executing the task on the MEC server or local in the local area for computation (line no: 118).”. It should be noted that edge cannot survive without cloud and there is strong dependency between device, edge and cloud. For some of the tasks/services the edge will always approach the cloud. The authors are suggested to revise the sentence.
- In Simulation settings, what is mean by Pass loss exponent? Do you mean Path loss exponent? Clarify it. Moreover, relevant references should be provided for some simulation settings such as distance between UEs and MEC servers, and distance between MEC servers. Why such kind of values have been selected? Is that based on some empirical study? Justify it with a relevant reference or your own empirical results.
- The reviewer cannot convince himself about the motivation of the paper. The motivation part of the paper is missing, and the proposed solution is too general. What would be an application scenario for the proposed work? The system model and the application model has been well articulated, however, how the proposed methods are benefiting any specific application? Authors are suggested to provide a section for motivation of their proposed work and explain it with a relevant killer application.
- Most of the literature in the paper is out of date. The references are not up to date and doesn’t cover it reasonably. Authors are suggested to add latest work from ACM/IEEE SEC, IEEE EDGE, HotEdge, FGCS, IEEE IoT, IEEE ICFEC. Some relevant reference should be added as follows:
- DOI: 1109/ACCESS.2018.2884536
- https://doi.org/10.1016/j.future.2020.04.033
- https://doi.org/10.1016/j.jnca.2020.102785
- DOI: 10.1109/ACCESS.2019.2914067
- The quality of images in the paper are not good. Please enhance the presentation quality of the figures.
- Conclusion usually summarizes the results, highlight achievements, and highlight possible applications and implications of the proposed work. I would suggest authors to add one more paragraph and conclude it with possible applications and implications and extend the future plan a bit more.
- I spotted a few typos as I was reading through the paper. For example, there should be a space between period and we (Line no 385). Moreover, acronyms should be spelled out upon first use, followed by the acronym itself in parentheses. For instance, the Internet of Things (IoT) has been used many times. Spell-out all the acronyms in the paper upon first use and then use throughout the paper.
Round 2
Reviewer 1 Report
From my point of view the paper is ready to be accepted
Reviewer 2 Report
The authors have significantly improved the manuscript considering all my comments. I believe the paper can be accepted in present form.